# How Can Machine Perfusion Change the Paradigm of Liver Transplantation for Patients with Perihilar Cholangiocarcinoma?

**DOI:** 10.3390/jcm12052026

**Published:** 2023-03-03

**Authors:** Damiano Patrono, Fabio Colli, Matteo Colangelo, Nicola De Stefano, Ana Lavinia Apostu, Elena Mazza, Silvia Catalano, Giorgia Rizza, Stefano Mirabella, Renato Romagnoli

**Affiliations:** General Surgery 2U—Liver Transplant Unit, Department of Surgical Sciences, Azienda Ospedaliero Universitaria Città della Salute e della Scienza di Torino, Università di Torino, Corso Bramante 88-90, 10126 Turin, Italy

**Keywords:** hilar cholangiocarcinoma, donor pool expansion, hypothermic oxygenated machine perfusion, normothermic machine perfusion, viability assessment, transplant oncology

## Abstract

Perihilar cholangiocarcinomas (pCCA) are rare yet aggressive tumors originating from the bile ducts. While surgery remains the mainstay of treatment, only a minority of patients are amenable to curative resection, and the prognosis of unresectable patients is dismal. The introduction of liver transplantation (LT) after neoadjuvant chemoradiation for unresectable pCCA in 1993 represented a major breakthrough, and it has been associated with 5-year survival rates consistently >50%. Despite these encouraging results, pCCA has remained a niche indication for LT, which is most likely due to the need for stringent candidate selection and the challenges in preoperative and surgical management. Machine perfusion (MP) has recently been reintroduced as an alternative to static cold storage to improve liver preservation from extended criteria donors. Aside from being associated with superior graft preservation, MP technology allows for the safe extension of preservation time and the testing of liver viability prior to implantation, which are characteristics that may be especially useful in the setting of LT for pCCA. This review summarizes current surgical strategies for pCCA treatment, with a focus on unmet needs that have contributed to the limited spread of LT for pCCA and how MP could be used in this setting, with a particular emphasis on the possibility of expanding the donor pool and improving transplant logistics.

## 1. Introduction

Perihilar cholangiocarcinomas (pCCA) are epithelial tumors originating from the biliary tree below second-order bile ducts and proximally to the confluence of the cystic duct, and they represent 50–70% of the tumors arising from the biliary tree [1]. They are relatively rare [2] but aggressive tumors, and surgical resection is generally considered the only potentially curative treatment [3,4]. However, most patients with pCCA are diagnosed at an advanced stage, and only 15–35% are amenable to curative resection [3,5,6], which is associated with a 15–40% 5-year survival [7,8,9]. The 5-year survival of patients suffering from unresectable pCCA is 2% [10].

The dismal prognosis of unresectable pCCA led to exploring liver transplantation (LT) following neoadjuvant treatment with external beam irradiation, brachytherapy, and 5-fluorouracil (5-FU) and/or oral capecitabine as a potential treatment. The first series from the Mayo Clinic reported an impressive intention-to-treat 54% 5-year survival and a 82% 5-year survival after transplantation [11]. However, although the survival benefit of this approach has been confirmed in subsequent series [12], LT for pCCA has not gained widespread acceptance due to the difficulties in applying the neoadjuvant protocol, patient selection and the lack of clear allocation rules in this setting. 

The term “transplant oncology” refers to the application of oncology along with transplant medicine and surgery to improve the survival and quality of life of cancer patients [13]. This includes considering LT for patients affected by malignancies that classically represented contraindications for LT, such as liver metastases from colorectal cancer [14], hepatocellular carcinoma beyond the most widely adopted selection criteria, pCCA and intrahepatic cholangiocarcinoma [15]. The prerequisite to successfully implement LT as a treatment for these diseases is the availability of suitable liver grafts. Although the introduction of direct acting antivirals against hepatitis C virus has profoundly changed the landscape of indications for LT [16], increasing the number of available grafts for alternative indications, the supply–demand gap for liver grafts remains an unresolved issue. The two main strategies to expand the donor pool are currently represented by the utilization of extended criteria donors (ECD) and by living donation. In most cases, ECD are represented by donors whose death has been determined by circulatory criteria (DCD), elderly donors, or liver grafts with significant macrovesicular steatosis [17,18]. While utilizations of these grafts may allow expanding the donor pool, their use has been associated with inferior outcomes as compared to those of LT using standard donors.

In the last decade, machine perfusion (MP) has been re-introduced in clinical practice, which is prompted by the need to cope with the increased risks associated with the use of ECD grafts [19,20,21,22,23,24,25,26,27,28,29,30,31,32,33,34,35,36,37,38,39,40,41,42,43,44,45,46,47,48,49,50,51,52,53,54]. Several MP techniques exist, which are characterized by different principles and mechanisms of graft protection [55]. Apart from improving graft preservation and allowing for longer preservation times, MP has a very interesting feature: it allows testing the viability of a liver graft prior to implantation (so-called “viability assessment”) [18,56]. Although normothermic MP (NMP) has been most frequently used as a tool for viability assessment, information about liver viability can be obtained also during hypothermic perfusion [33,57]. Assessing the viability of a graft should ideally allow for an increase in the number of transplanted grafts while minimizing recipient risk and avoiding discarding potentially usable grafts solely based on donor characteristics. In addition, other aspects of machine perfusion technology make its application in the setting of LT for pCCA appealing. 

This review will summarize some important aspects of pCCA surgical management, emphasizing the need to improve the oncologic outcome of both resectable and unresectable patients. Literature on the results of LT for pCCA will be reviewed, discussing the limitations of current approaches. Finally, potential applications of MP in pCCA treatment of will be reviewed. 

## 2. The Challenge of Perihilar Cholangiocarcinoma

As the international classification of cholangiocarcinoma does not distinguish between perihepatic and distal cholangiocarcinoma [4], estimating the true incidence of pCCA is difficult. In the West, age standardized incidence rates range between 0.5 and 2 per 100,000 individuals, whereas in eastern Asia, incidence is higher due to endemic liver flukes (*Opisthorchis viverrini* and *Clonorchis sinensis*) infection as well as a higher incidence of hepatolithiasis. Worldwide, the incidence of pCCA has increased in recent years, which has been linked to the increased incidence of metabolic syndrome, especially in countries with historically low incidence rates [2]. 

Perihilar CCA is an aggressive disease. A large study from the Netherlands on 2031 patients showed an overall median survival of 5.2 months [58]. Patients undergoing palliative systemic treatment, loco-regional treatment or best supportive care had a median survival of 12.2, 14.5 and 2.9 months, respectively. Notably, only 15% of patients underwent curative resection, which was associated with a median survival of 29.6 months [58]. 

### 2.1. Surgery for Perihilar Cholangiocarcinoma

The outcome of patients suffering from pCCA is primarily determined by the possibility to undergo curative resection. However, only a minority of patients are eligible for surgical resection due to several factors. Early diagnosis is infrequent in pCCA because most patients with early disease are asymptomatic or symptoms are poorly specific (dyspepsia, abdominal discomfort, fatigue, weight loss) [3]. Furthermore, pCCA are desmoplastic and paucicellular tumors, which complicates obtaining histological confirmation once the clinical diagnosis becomes more evident [59]. At this stage, most patients will present with jaundice and/or cholangitis and will frequently require preoperative biliary drainage (PBD). In patients undergoing surgery for pCCA, preoperative cholangitis is associated with increased mortality, overall morbidity, incidence of liver failure, and sepsis, and it is an absolute indication for PBD [60]. In patients with jaundice but not cholangitis, PBD is still frequently indicated due to the concerns for impaired liver regeneration capability, as pCCA patients are frequently candidate for major liver resections. However, PBD has been associated with higher overall morbidity, perioperative transfusion, cholangitis, infection and bile leakage [61,62], suggesting that it could be reasonably avoided in patients with sufficient future liver remnant (≥50%). It is significant that regardless of the technique used for PBD (endoscopic versus percutaneous transhepatic biliary drainage), about 15% of patients will fail to proceed to surgery because of PBD complications and progressive deterioration [63]. Another factor complicating the surgical approach is the necessity to perform an oncologically adequate (R0) surgery, which frequently involves an extended hepatectomy associated with the resection of the biliary confluence and the reconstruction by an hepaticojejunostomy while preserving a sufficient portion of liver parenchyma. Portal vein embolization has traditionally been used to induce future liver remnant hypertrophy. Associating liver partition and portal vein ligation for stage hepatectomy (ALPPS) represents an alternative approach [64]. However, ALPPS is still debated in the setting of pCCA [65,66]. In patients who do not develop sufficient liver hypertrophy after portal vein embolization alone, associating hepatic vein embolization (so-called liver venous deprivation) could contribute to enhancing the growth of future liver remnants and improve access to curative resection [67].

Patients who can access resection with curative intent are exposed to an overall major morbidity rate of 43–65%, whereas postoperative mortality rates as high as 17% have been reported [68,69]. In a study evaluating outcomes of pCCA resection in 708 low-risk patients at 24 high-volume centers, the benchmark values (i.e., the 75% or 25% percentiles of the medians of each center) for Clavien–Dindo ≥ 3 complications rate and in-hospital mortality were ≤70% and ≤8%, respectively [70]. 

About 80% of patients will experience recurrence after resection, in most cases within 2 years from surgery [71,72]. Overall 5-year survival is 11–44% and appears to be strongly influenced by the radicality of surgical resection, being ~60% in patients undergoing R0 resection versus <10% after R1 resection [69]. Interestingly, benchmark value for R1 resection has been set at ≤43% [70]. 

Overall, surgery with curative intent appears to be an option only in a minority of patients suffering from pCCA, and it is burdened by a complicated preoperative management, high postoperative morbidity and mortality, and high recurrence rates, which highlights the urgent need for alternative strategies to improve the outcome in these patients. 

### 2.2. Liver Transplantation as a Treatment for Perihilar Cholangiocarcinoma

In theory, LT is an interesting option for patients with pCCA because it allows for the radical excision of the tumor while avoiding the issue of residual hepatic functional reserve. Unfortunately, early results of LT performed in patients with pCCA were burdened by high recurrence rates, leading to pCCA being considered a contraindication for LT. [73,74]. However, observations that long-term survival could be achieved in patients with limited tumor burden, negative resection margins and no lymph node involvement opened to reconsider pCCA as a possible indication for LT in selected patients [75]. As aforementioned, the early experiences from the Mayo Clinic (Rochester, MN, USA) team showed that by stringent patient selection and by applying a neoadjuvant protocol of external beam radiotherapy, brachytherapy and 5-FU, excellent results could be achieved [11,76,77]. Table 1 summarizes the results of LT for pCCA [11,12,76,78,79,80,81,82,83,84,85,86,87,88,89,90]. 

In the absence of a neoadjuvant protocol, LT has been associated with 5-year overall survival rates ranging from 20% to 36%, whereas using a pre-transplant chemoradiation protocol has resulted in 5-year survival rates ranging from 52% to 82%. These positive outcomes have come at the expense of strict patient selection and the morbidity of the neoadjuvant treatment itself. Indeed, 25–42% of patients initially candidate to LT after chemoradiation will not be transplanted due to inability to tolerate the treatment, complications, or tumor progression. Furthermore, LT can be technically complicated due to the effects of radiotherapy on the hepatic hilum. Since the early reports [11], an increased incidence of hepatic artery and portal vein thrombosis has been reported, leading to the frequent choice of utilizing an interposition graft anastomosed to infrarenal aorta for arterial vascularization. Early postoperative outcomes have been marked by a higher rate of complications, sometimes directly related to preoperative radiation therapy. Another element of difficulty may be represented by the presence of adhesions. Indeed, a staging laparotomy is indicated to rule out peritoneal disease or extrahepatic lymphnodes involvement before the patient can be considered eligible for LT. In the setting of deceased donor LT, considerable time can separate the staging laparotomy from LT operation, further complicating an already difficult dissection. An alternative option, which has been adopted by some centers, is performing the staging laparotomy simultaneously with LT, to avoid a repeat operation and peritoneal adhesions. While this is a viable option in living donor liver transplantation, in deceased donor LT, it necessitates the availability of a back-up recipient and has the disadvantage of significantly prolonging preservation time, which may have a negative impact on postoperative graft function.

In summary, although excellent outcomes have been reported, LT for pCCA has not gained widespread adoption. This is likely explained by the limited number of eligible patients, the difficulties in preoperative management and the technical and logistical difficulties linked to the neoadjuvant chemoradiation protocol.

## 3. Machine Perfusion in Liver Transplantation for Perihilar Cholangiocarcinoma: A Game Changer?

Based on the available evidence, there are two aspects of MP technology that could be of particular interest in the setting of LT for pCCA: (1) the possibility of expanding donor pool by improving the preservation of grafts from ECD and by testing their viability; (2) the possibility of improving transplant logistic by prolonging preservation time (Figure 1). 

### 3.1. Expanding Donor Pool and Viability Assessment

In the Italian liver allocation system, patients with pCCA can benefit from a priority allocation based on a multidisciplinary discussion involving transplant surgeons, hepatologists and intensive care anesthetists [94]. It has been proposed that 5% of organ donor pool could be allocated to novel indications for which strong scientific evidence is lacking. However, in a system already stressed by a chronic organ donor shortage, this may be difficult to achieve. Campaigns promoting organ donation, the use of extended criteria donors, and living donations are all effective ways to increase the donor pool.

Despite the widespread gap between organ demand and supply, there is a significant disparity in many countries between the number of offered organs and the number of those that are eventually transplanted. In 2021, 23% of signaled livers in Italy were not transplanted because of general contraindications to organ donation or because they were judged unsuitable for LT. The situation is similar in other countries, such as the UK [29] or USA [95]. Traditionally, the choice of accepting an organ offer has been based on donor and recipient characteristics. Appropriately weighing the risk profile associated with each donor–recipient match is a fine art, and many scores have been proposed to help transplant surgeons make the difficult decision of accepting an organ for a specific recipient [96,97,98].

Machine perfusion is associated with a significant reduction in ischemia–reperfusion injury associated with LT, as demonstrated by randomized controlled trials [20,24,31,38,45,99] and retrospective studies [19,21,22,25,26,32,34,35,36,37,40,42,43,44,47,48,95,100,101,102,103,104,105,106,107]. Although clinical indication for its use is still heterogeneous [108], many groups have now implemented this technology into routine clinical practice and others are enthusiastically starting to adopt it [109]. Implementing MP technology can effectively lead to a donor pool expansion by changing the perceived risk profile associated with a specific organ offer and allowing for the successful use of a greater number of ECD grafts. As a result, transplant professionals may be more willing to consider higher-risk offers and to use ECD grafts that would otherwise be discarded.

However, this decision would still be based on a presumed risk, similarly to what happens when the graft is preserved by static cold storage. The possibility of testing liver viability during preservation challenges this concept. Indeed, one fundamental aspect of MP is the possibility to assess the function and metabolism of the liver to be transplanted ex situ, after the damage sustained during procurement and initial cold preservation. MP represents an unbiased environment, in which objective parameters guiding graft acceptance can be gathered [110,111]. While this property has classically been referred to normothermic machine perfusion [29,54,112,113], recent studies suggest that precious information about liver viability and post-LT can be obtained also during cold preservation [33,57,114]. The most widely adopted criteria for viability assessment during normothermic MP are based on lactate and glucose metabolism, pH homeostasis, vascular flows, perfusate transaminases and bile production and composition. However, at least in theory, any metabolic function can be tested during MP and serve as a further element to assess liver viability. When used on livers that were previously deemed unsuitable for LT, NMP has allowed for the successful transplantation of 46% to 100% of them, confirming its enormous potential in expanding the donor pool. However, the primary non-function of normothermic MP-treated livers has been reported [48,115]. Furthermore, normothermic MP, especially when applied after a period of cold preservation (so-called “back-to-base” approach) has been shown to be suboptimal in preventing the development of non-anastomotic biliary strictures [29]. As a result, some centers have included parameters to assess cholangiocyte viability in their protocols, which has improved the ability to predict the development of ischemic cholangiopathy. These criteria have been criticized as they might be too restrictive, and the debate about how high-risk livers should be evaluated during NMP is still ongoing. Evaluation protocols are heterogeneous and constantly evolving. While an element of subjectivity in the complex decision of accepting a liver graft appears to be unavoidable, MP appears to have enormous potential for increasing ECD utilization and expanding donor pool, thereby improving access to LT for patients suffering from pCCA. 

Table 2 summarized studies on liver viability assessment during MP. 

### 3.2. Improving Transplant Logistics

Time is a critical issue in LT for pCCA. The transplanting surgeon must deal with several issues at once, including adhesions from the previous staging laparotomy, fibrosis and tissue thickening from radiation therapy at the hepatic hilum, and the frequent need to perform a complex hepatic artery reconstruction using an interposition graft anastomosed to the abdominal aorta. In patients suffering from primary sclerosing cholangitis, concomitant portal hypertension may further complicate LT operation. One option for avoiding adhesions is to perform lymphnode sampling concurrently with LT; however, even in this case, the time required to confirm the absence of lymphnode involvement may prolong preservation time. Additionally, in the unfortunate case of lymph node involvement, preservation time would become prohibitively long, exposing the back-up recipient to a high risk of post-LT graft dysfunction. While liver grafts from optimal donors may tolerate longer preservation time, those from ECD are more susceptible to severe ischemia–reperfusion injury when cold ischemia time is prolonged.

MP technology can be used to safely prolong preservation time [46,54,101,112,116], possibly facilitating transplant organization and transforming it into a semi-elective procedure. In the randomized controlled trial by Nasralla et al. [31], improved postoperative outcomes were observed despite significantly longer preservation time in the MP group. With regard to normothermic MP, undoubtedly, the experience from the Innsbruck group represents a model of organization [101]. At this center, the liver is handed over to the intensive care unit team after it has been connected to the MP device by the on-call surgeon. In the ICU, the liver is monitored like a patient. At the end of the preservation, device and perfusate parameters are reviewed and, if the liver is confirmed as transplantable, LT is scheduled. The positive impact of NMP on transplant logistics has also been highlighted by the Birmingham group in the recently published NAPLES study [117]. In this study, outcomes of repeat LT performed using NMP-preserved suboptimal liver grafts were compared to those performed with optimal livers preserved by static cold storage. As the outcomes in both cohorts were comparable, the authors concluded that NMP enabled them to achieve comparable outcomes despite using grafts from extended-criteria donors, thereby improving access to LT. Another important aspect of MP in repeat LT is the possibility of relieving time pressure from the transplanting surgeon having to perform a difficult recipient hepatectomy. This approach, which can be applied also based on logistical aspects and recipient characteristics, could be of value in the setting of pCCA. The ability to extend preservation time without compromising post-LT graft function could allow the transplanting surgeon to perform an accurate lymphadenectomy, wait for the pathologist’s response, and then proceed with a difficult hepatectomy and complex arterial reconstruction. Obviously, given the time constraints of LT for pCCA, it would be unwise to abuse MP technology and begin LT with an already extended preservation time with the risk of discarding the graft should the recipient be ultimately not transplantable. 

It should be noted that the possibility of prolonging preservation time is not exclusive of NMP. A recent multicenter European study has highlighted that a period of hypothermic oxygenated machine perfusion ≥ 4 h has no detrimental consequences for graft function and patient outcome [118]. Based on these results, the Groningen group designed a randomized controlled trial to test the safety of pronged hypothermic oxygenated MP, in which livers procured after 4 p.m. and 4 a.m. will be treated with prolonged dual hypothermic oxygenated machine perfusion and transplanted the following day [119]. The trial has completed recruitment, and results are expected soon [109]. 

It is worth noting that some of the benefits of MP, particularly the ability to extend preservation time, could potentially apply to other indications of transplant oncology, such as LT for colorectal cancer hepatic metastases, where patients frequently undergo LT after repeated hepatic resections and recipient hepatectomy can be challenging.

## 4. Conclusions

LT represents a potentially curative treatment for patients suffering from pCCA. The reported outcomes of LT in this setting, which compare favorably to surgical resection, have raised the question of whether LT should be offered to selected patients with resectable disease [120]. MP technology could help overcome some of the obstacles complicating this approach.

Thanks to a better understanding of genetics and molecular biology [4,121], it is likely that in the upcoming years, the armamentarium of treatments for intrahepatic and perihilar cholangiocarcinoma will expand significantly [4,122,123]. Hopefully, this will result in a greater number of previously unresectable patients becoming eligible for surgical resection or LT. To be sustainable, any expansion of the indications for LT must be accompanied by an increase in the number of available grafts. Even if more effective target- and immunotherapy will possibly avoid the need for neoadjuvant radiation and the complications related to this approach, the problem of organ supply will remain. In this view, MP will be instrumental in optimizing preservation of ECD livers and allowing a safe donor pool expansion.

## Figures and Tables

**Figure 1 jcm-12-02026-f001:**
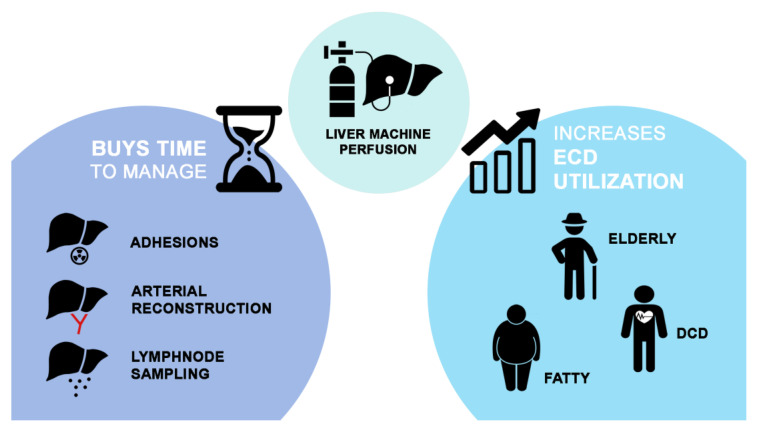
A visual representation of the possible advantages of MP technology in the setting of liver transplantation for perihilar cholangiocarcinoma.

**Table 1 jcm-12-02026-t001:** Results of LT for pCCA.

Author	Country	Study Design	n	Dropout (%)	Neoadjuvant Treatment	Survival Outcomes
Figueras et al. [83]	Spain	Single center, retrospective	LT, n = 8 LR, n = 20	n.a.	None	5-year survival: - LT = 36% - Resection = 21%
Sudan et al. [91]	NE, USA	Single center, retrospective	LT, n = 11	35%	Brachytherapy 6000 cGy + 5-FU	Median survival after LT = 25 months; 45% disease-free with median 7.5 years follow-up
Heimbach et al. [11]	MN, USA	Single center, prospective	LT, n = 28	39%	EBRT 4500 cGy + Brachytherapy 2000-3000 cGy + 5-FU	5-year survival: - Whole cohort = 54% - LT = 82%
Robles et al. [88]	Spain	Multicenter, retrospective	LT, n = 36	n.a.	None	Overall survival at 1, 3, 5, and 10 years was 82%, 53%, 30%, and 18%. Disease-free survival at 1, 3, 5, and 10 years was 77%, 53%, 30%, and 18%.
Axelrod et al. [79]	IL, USA	Single center, retrospective	LT, n = 5	n.a.	EBRT 45 Gy + 5-FU	100% recurrence-free survival in 4 patients treated with neoadjuvant protocol (median follow-up = 18 months)
Jonas et al. [85]	Germany	Single center, retrospective	LT, n = 5	n.a.	None	Overall survival was 80% at a median follow-up of 20 months
Hidalgo et al. [84]	UK	Single center, retrospective	LT, n = 12 LR, n = 44	n.a.	None	5-year survival: - LT = 20% - Resection = 28%
Kaiser et al. [86]	Germany	Multicenter, retrospective	LT, n = 47	n.a.	None	Median survival = 35.5 months. Overall survival at 1, 3 and 5 years was 61%, 31% and 22%
Rosen et al. [92]	MN, USA	Single center, retrospective	LT, n = 136	36%	EBRT 4500 cGy + Brachytherapy 2000-3000 cGy + 5-FU + capecitabile	Overall survival at 1, 3 and 5 years was 92%, 81%, and 74%.
Darwish Murad et al. [12]	USA	Multicenter, retrospective	LT, n = 214	25%	EBRT 4500 cGy + Brachytherapy 2000-3000 cGy + 5-FU + capecitabile	Recurrence-free survival at 2, 5 and 10 years was 78%, 65% and 59%.
Schule et al. [89]	Germany	Single center, retrospective	LT, n = 16	n.a.	None	Overall survival (postoperative deaths excluded) at 3 and 5 years was 63% and 50% in N0 patients and 15% and 0% in N+ patients
Welling et al. [93]	USA	Single center, retrospective	LT, n = 6	42%	SBRT 50-60 Gy + capecitabine	Overall survival in transplanted patients at 1 year was 81%
Duignan et al. [81]	Ireland	Single center, retrospective	LT, n = 20	26%	EBRT 45-55 Gy + Brachytherapy 7.5 Gy + 5-FU + capecitabile	Overall survival at 1, 3 and 4 years was 75%, 60% and 51%
Marchan et al. [87]	GA, USA	Single center, retrospective	LT, n = 8	20%	EBRT 4500 cGy + Brachytherapy 2000-3000 cGy + 5-FU + capecitabile	Median survival = 30.2 months. Overall survival at 6, 12 and 24 months was 100%, 87.5%, and 87.5%
Dondorf et al. [80]	Germany	Single center, retrospective	LT, n = 22	31%	None	Median survival = 29 months. Overall survival at 1, 3 and 5 years was 89,2%, 36% and 28.8%.
Ethun et al. [82]	USA	Multicenter, retrospective	LT, n = 41 LR, n = 191	34%	EBRT 4500 cGy + Brachytherapy 2000-3000 cGy + 5-FU	Median survival: - LT = 77.4 months - Resection = 27 months
Zaborowski et al. [90]	Ireland	Multicenter, retrospective	LT, n = 26	30%	EBRT 45-55 Gy + Brachytherapy 7.5 Gy + 5-FU + capecitabile	Median survival = 53 months. Overall survival at 1, 3 and 5 years was 81%, 69% and 55%.
Ahmed et al. [78]	MO, USA	Single center, retrospective	LT, n = 38	34%	EBRT 4500 cGy + Brachytherapy 2000-3000 cGy + 5-FU	Overall survival at 1, 3 and 5 years was 91%, 58% and 52%

Abbreviations: LT, liver transplantation; LR, liver resection; EBRT, external beam radiotherapy; SBRT, stereotactic beam radiotherapy; 5-FU, 5-fluoruracil.

**Table 2 jcm-12-02026-t002:** Studies on liver viability assessment during MP.

Author	n	DCD	Time	Viability Criteria	Utilization Rate
Viability assessment during normothermic MP					
Mergental et al. 2016 [30]	6	4/5 (80%)	3 h	Perfusate lactate level < 2.5 mmol/L or evidence of bile production + at least 2 of the following: (1) pH > 7.3; (2) stable vascular flows (hepatic artery flow > 150 mL/min and portal vein flow > 500 mL/min; (3) homogeneous perfusion and soft consistency	5/6 (83.3%)
Watson et al. 2017 [51]	12	9/12 (75%)	n.a.	Changes in perfusate lactate, glucose and transaminases concentration + ability to maintain pH without supplemental bicarbonate	n.a.
Watson et al. 2018 [48]	47	35/47 (74.5%)	≤6 h	Variables associated with successful transplantation: (1) Maximum bile pH > 7.5; (2) bile glucose concentration ≤ 3 mmol/L or ≥10 mmol less than perfusate glucose; (3) ability to maintain perfusate pH > 7.2 with ≤30 mmol bicarbonate supplementation; (4) falling glucose beyond 2 h or perfusate glucose under 10 mmol/L which, on challenge with 2.5 g glucose, does subsequently fall; (5) peak lactate fall ≥ 4.4 mmol/L/kg/h; (6) Perfusate ALT < 6000 IU/L at 2 h	22/47 (46.8%)
de Vries et al. 2019 [49]	7	7/7 (100%)	2.5 h	All of the following: (1) lactate < 1.7 mmol/L; (2) perfusate pH 7.35 to 7.45; (3) bile production > 10 mL; (4) biliary pH > 7.45	5/7 (71.4%)
Matton et al. 2019 [50]	6	6/6 (100%)	2.5 h	(1) Biliary bicarbonate > 18 mmol/L; (2) biliary pH > 7.48; (3) biliary glucose < 16 mmol/L; (4) bile/perfusate glucose concentration ratio < 0.67; (5) biliary LDH < 3689 IU/L	4/6 (66.7%)
van Leeuwen et al. 2020 [43]	16	16/16 (100%)	2.5 h	All of the following: (1) lactate < 1.7 mmol/L; (2) perfusate pH 7.35 to 7.45; (3) bile production > 10 mL; (4) biliary pH > 7.45	11/16 (68.7%)
Mergental et al. 2020 [29]	31	14/31 (45.2%)	4 h	Perfusate lactate level < 2.5 mmol/L or evidence of bile production + at least 2 of the following: (1) pH > 7.3; (2) stable vascular flows (hepatic artery flow > 150 mL/min and portal vein flow > 500 mL/min; (3) homogeneous perfusion and soft consistency	22/31 (71%)
Reiling et al. 2020 [52]	10	5/10 (50%)	2–4 h	(1) Lactate clearance to <2 mmol/L within 2 h; (2) glucose metabolism as evidenced by a decreasing trend in serum glucose concentration by 4 h; (3) maintenance of physiological pH; (4) stable hepatic arterial and portal venous flows; (5) homogeneous graft perfusion with soft consistency of parenchyma; (6) bile production (no lower limit)	10/10 (100%)
Hann et al. 2021 [53]	5	0/5 (0%)	6 h	Perfusate lactate level < 2.5 mmol/L or evidence of bile production + at least 2 of the following: (1) pH > 7.3; (2) stable vascular flows (hepatic artery flow > 150 mL/min and portal vein flow > 500 mL/min; (3) homogeneous perfusion and soft consistency	n.a.
Quintini et al. 2022 [37]	21	13/21 (61.9%)	6 h	At least two of the following: (1) lowest perfusate lactate level <4.5 mmol/L or a decrease of 60% from peak in the first 4 h; (2) bile production rate higher than 2 mL/h; (3) stable HA flow of >0.05 mL/min/g of liver weight and PV flow >0.4 mL/min/g of liver weight; (4) macroscopic homogenous perfusion and soft consistency.	15/21 (71.5%)
van Leeuwen et al. 2022 [42]	54	53/54 (98.2%)	2.5 h	“Green zone” criteria: (1) lactate < 1.7 mmol/L; (2) perfusate pH 7.35 to 7.45; (3) bile production > 10 mL; (4) biliary pH > 7.45; (5) Δ pH > 0.10; (6) Δ HCO3- > 5 mmol/L; (7) Δ glucose < −5 mmol/L	34/54 (63%)
Watson et al. 2022 [47]	203	123/203 (61%)	2 h	ALT, lactate, supplementary bicarbonate (first 4 h), and peak bile pH associated with early allograft function	154/203 (76%)
Clavien et al. 2022 [54]	1	0/1 (0%)	3 d	Physiologic response to hormones and vasoactive drugs, quality and quantity of bile production, histology, significant decline in injury (AST/ALT, uric acid) and inflammation (IL-6) markers in the perfusate	1/1 (100%)
Viability assessment during hypothermic oxygenated MP					
Muller et al. 2019 [57]	54	35/54 (65%)	30 min	Perfusate flavin mononucleotide (threshold 100 ng/mL) correlated with postoperative graft function, complications and graft survival	n.a.
Patrono et al. 2020 [33]	50	0/50 (0%)	2 h	Perfusate parameters (pH, glucose, lactate, AST, ALT and LDH) correlated with early allograft dysfunction, with ALT performing best. Macrovesicular steatosis was the only factor independently associated with postoperative graft function	n.a.

Abbreviations: DCD, donation after circulatory death; PNF, primary non-function; IC, ischemic cholangiopathy; AST, aspartate aminotransferase; ALT, alanine aminotransferase; LDH, lactate dehydrogenase; d, days; h, hours; min, minutes.

## Data Availability

Not applicable.

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
