# Peer review of "How Can Machine Perfusion Change the Paradigm of Liver Transplantation for Patients with Perihilar Cholangiocarcinoma?"

_jcm, 2023, doi:10.3390/jcm12052026_

Round 1

Reviewer 1 Report

This is a well-written clear review article, albeit over a very small indication area. pCCA is rare, and patients eligible for LT are even rarer. It is questionable whether machine perfusion is a game changer here in particular. Especially in the event that a patient is not transplantable after all, it would be very annoying if the CIT with or without MP was already far advanced and the organ could therefore not be further allocated. Therefore, I would not advocate using the MP to transplant a pCCA patient next morning. Please consider this in the discussion.

Author Response

Thank you for your time and remarks.

With regards to your first point, we agree that nowadays patients with pCCA potentially amenable to LT are a minority, but we do believe that the scarce consideration of LT as a treatment for pCCA issues from its numerous logistical hurdles, some of which MP could help to overcome. The introduction of new target therapies has the potential for rescuing patients initially deemed not eligible for resection or LT, and will hopefully allow for a moderate expansion of this indication. We acknowledge this is somewhat speculative, so we double-checked the manuscript to ensure the appropriate tenses have been used. We also stressed (heading 3.2, last paragraph) that some advantages of MP, especially prolonging preservation time, could potentially apply to other indications, like liver transplantation for colorectal cancer hepatic metastases, in which patients often undergo LT after repeated hepatic resections and recipient hepatectomy can be challenging.

Concerning your second point, we absolutely agree that discarding a graft due to excessive preservation time should be avoided by any means. We do not mean to encourage the use of MP to schedule LT during daytime in pCCA. We just mentioned this possibility to corroborate the fact that NMP does allow safely extending preservation time. To avoid misunderstandings, we have added the following sentence (heading 3.2, end of 2nd paragraph): “Obviously, given the time constraints of LT for pCCA, it would be unwise to abuse of MP technology and begin LT with an already extended preservation time, with the risk of discarding the graft should the recipient be ultimately not transplantable.”

Reviewer 2 Report

thank you for writing this review. The use of machine perfusion as modality for viability assessment is a hot topic in transplant medicine but it is important that other specialties also read about its opportunities. Solving donor organ shortage is an important goal to help every patient that has end-stage organ failure. It would be wonderful if a transplantation can help cancer patients. 

The readability of the paper is quite poor due to long and difficult sentences nut also the use of wrong english words/terms and also the structure of sentences. I would advise to let the paper be checked by a native english speaker. 

Author Response

Thank you for your appreciation and for capturing the very purpose of our paper. Following your suggestion, the manuscript has been extensively edited for fluency and brevity with the help of a native English speaker. Inappropriate wording has been fixed and long sentences have been reformulated to improve clarity. In case additional concerns remain, we will be glad to try to meet them upon detailed specification.